# Shallow very-low-frequency earthquakes accompany slow slip events in the Nankai subduction zone

Masaru Nakano [1], Takane Hori[1], Eiichiro Araki[1], Shuichi Kodaira[1] & Satoshi Ide[2]

Recent studies of slow earthquakes along plate boundaries have shown that tectonic tremor, low-frequency earthquakes, very-low-frequency events (VLFEs), and slow-slip events (SSEs) often accompany each other and appear to share common source faults. However, the source processes of slow events occurring in the shallow part of plate boundaries are not well known because seismic observations have been limited to land-based stations, which offer poor resolution beneath offshore plate boundaries. Here we use data obtained from seafloor observation networks in the Nankai trough, southwest of Japan, to investigate shallow VLFEs in detail. Coincident with the VLFE activity, signals indicative of shallow SSEs were detected by geodetic observations at seafloor borehole observatories in the same region. We find that the shallow VLFEs and SSEs share common source regions and almost identical time histories of moment release. We conclude that these slow events arise from the same fault slip and that VLFEs represent relatively high-frequency fluctuations of slip during SSEs.

[1] R&D Center for Earthquake and Tsunami, Japan Agency for Marine-Earth Science and Technology, 3173-25 Showa-machi, Kanazawa, Yokohama 236-0001, Japan. [2] Department of Earth and Planetary Science, University of Tokyo, 7-3-1 Hongo, Bunkyo, Tokyo 113-0033, Japan. Correspondence and requests for materials should be addressed to M.N. (email: mnakano@jamstec.go.jp)

Recent seismic and geodetic observations have revealed the existence of a family of slow earthquakes occurring along or near plate boundaries around the world[1–3]. Slow earthquakes can be classified into several types, including low-frequency or tectonic tremor ('tremor' hereafter) and impulsive low-frequency earthquakes (LFEs) with frequencies around several hertz[1, 4], very-low-frequency events (VLFEs) with periods of 10–100 s[5, 6], and slow-slip events (SSE) with durations of days to years[7, 8]. Because of their importance for understanding slip behavior along faults, especially for plate boundary faults, slow earthquakes have been intensively studied by many researchers to explore several possibilities. For instance, the sensitivities of slow earthquakes to stress change may function as stress meters along subduction zones[3]. Slow earthquakes near the shallowest part of the plate boundary would share source regions with tsunami earthquakes[9]. And comprehensive studies of regular and slow earthquakes can improve our understanding of earthquake generation mechanisms in ways that may help disaster mitigation from megathrust earthquakes. However, slow events occurring offshore, especially beneath the toe of accretionary prisms at subduction plate boundaries, are poorly understood because observation networks of land-based stations are at a disadvantage in detecting their weak signals. Data from seafloor observation networks, now being put into place, are necessary to improve our understanding of faulting characteristics of offshore earthquakes of all kinds[10].

Slow earthquakes have been intensively studied along the Nankai trough, southwest of Japan, where the Philippine Sea plate is subducting beneath the Amur plate. Megathrust earthquakes of magnitude ($M_w$) 8 or larger have repeatedly occurred in the Nankai trough at intervals of 100 to 150 years[11, 12]. Slow earthquakes occur on deeper and shallower extensions of the source regions of these earthquakes. Previous studies have revealed that slow earthquakes are often accompanied by other types of slow earthquakes in the same source region; for instance, deep SSEs are often accompanied by tremor[13–17], and deep tremor has been reported to accompany deep VLFEs[6, 18–20]. The correlation of tremor and VLFEs has also been observed in Mexico, New Zealand, and Cascadia[21–23]. Observations using ocean-bottom seismographs have shown that signals of shallow VLFEs (sVLFEs) overlap tremor signals[9, 24]. The incidence of sVLFEs has been found in some cases to be modulated by SSEs occurring on adjacent faults[17, 25–27]. It also has been inferred that sVLFEs accompany shallow SSEs[24, 26], but shallow SSEs are not well documented given the difficulty of detecting them. Given these spatial and temporal correlations among slow earthquakes, source models have been proposed to explain variations in their radiated frequency ranges and waveform characteristics[28–30]. However, some SSEs or VLFEs have been observed without other types of slow slip[31, 32], indicating that slow earthquakes have complicated source processes.

Recent observations from seafloor borehole stations have documented repeated changes in formation of pore pressure, which were responses to magnitude 5 class SSEs along the Nankai trough plate boundary beneath the toe of the accretionary prism, accompanying tremor[33]. In this paper, we report quantitative correlations of sVLFEs and shallow SSEs in the Nankai trough, indicating that these slow events share a common source fault and slip.

## Results

**sVLFEs along the Nankai trough.** Along the full length of the Nankai trough, sVLFEs have been repeatedly detected[5, 9, 10, 26, 27, 34–37]. Off the Kii Peninsula, in the eastern part of the Nankai trough, the distribution of sVLFEs is sporadic (Fig. 1) in contrast with the continuous distribution of tremor[1] and VLFEs[20] occurring along the deeper extension of the subducting plate in this region. The activity of sVLFE is intermittent and continues for several weeks to several months. The sVLFE

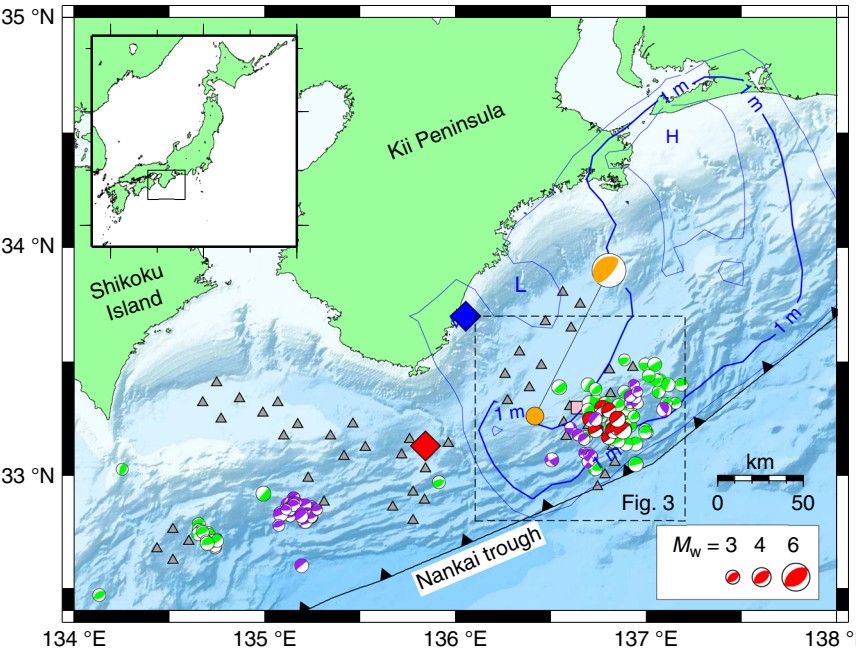

**Fig. 1** Locations of the observation network and sVLFEs along the Nankai trough. Gray triangles indicate DONET stations; the pink square near the center of the dashed rectangle is borehole station C0002G. Focal mechanism symbols indicate sVLFEs in 2003 and 2004[34] (green), 2009[9] (red), and 2015[10] (purple). The orange circle with corresponding focal mechanism symbol is the epicenter of the 1 April 2016 off-Mie earthquake ($M_{JMA}$ 6.5); blue and red diamonds are the epicenters of the 1944 Tonankai and 1946 Nankai earthquakes, respectively. Blue contours represent the slip distribution during the 1944 Tonankai earthquake[43] (contour interval 0.5 m). L and H mark regions of low and high slip, respectively

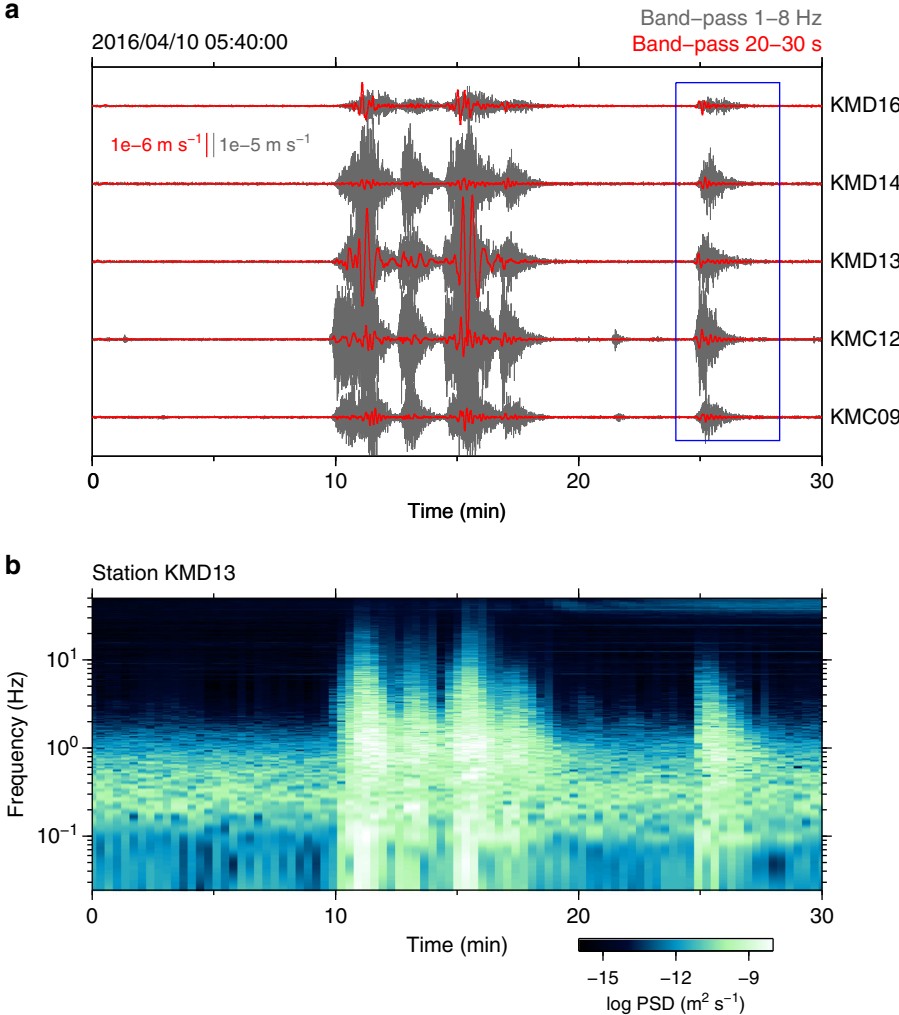

**Fig. 2** An example of sVLFE burst activity in vertical component seismograms. **a** Gray and red waveforms represent short-period (band-passed 1–8 Hz) and long-period (band-passed 20–30 s) components, respectively. The blue rectangle outlines the sVLFE for which the CMT inversion result is shown in Supplementary Fig. 2. **b** Running spectrum of the waveform observed at station KMD13, the middle waveform in **a**

episodes in September 2004 and in 2011 appear to have been triggered by large earthquakes, such as the 2004 off-Kii Peninsula earthquake or the 2011 Tohoku-oki earthquake[5, 37], whereas other episodes in 2003, May 2004, 2009, and 2015 had no known triggering events[5, 9, 10].

Another burst of sVLFE activity occurred in 2016, apparently triggered by the off southeast Mie Prefecture earthquake ($M_{JMA}$ 6.5) of 1 April 2016 ('off-Mie earthquake' hereafter) that ruptured the Nankai trough plate boundary beneath the accretionary prism[38] (Fig. 1), and shallow SSEs and shallow tremor were also observed after this event[33]. The sVLFE activity started about 1 h after the mainshock, but the majority of events occurred between 2 and 17 days after the mainshock. Figure 2 shows an example of a burst of sVLFE activity that occurred during this period. The observed amplitude spectrum was mostly flat at frequencies lower than 1 Hz[39].

**Common fault slip of sVLFEs and SSEs.** We determined the centroid moment tensors (CMTs) of the sVLFEs in the 2016 episode (see Methods) by using data from the Dense Oceanfloor Network System for Earthquakes and Tsunamis (DONET) installed along the Nankai trough (Fig. 1 and Supplementary Fig. 1)[40, 41]. Most of the events had thrust-type focal mechanisms (Supplementary Fig. 2) and were distributed about 20 km updip

from the mainshock of the off-Mie earthquake at depths of 6 to 9 km below sea level, in an oval region measuring ~30 km by 50 km with its long axis parallel to the trough (Fig. 3 and Supplementary Fig. 3). Several events with strike-slip focal mechanisms were located deeper than 10 km, but their location errors were larger than those for the other events. Although these events might emerge from different source locations or processes, almost all of other sVLFEs represent low-angle thrust movement on the plate interface. The sVLFE episode of 2016 overlaps the area of the 2009 episode and the western cluster of the 2015 episode. The eastern part of the 2015 episode had no distinct activity in 2016, although that area also saw sVLFE activity in 2004 (Fig. 1). The overall distribution of sVLFE activity corresponds to an area of small slip deficit rate (weak coupling) along the plate boundary[42] and the area of large slip during the great 1944 Tonankai earthquake[43].

The temporal evolution of sVLFEs and aftershocks of the off-Mie earthquake were quite different (Fig. 4a). Aftershock activity mostly ceased by 3 April 2016, whereas sVLFEs began that day and continued until 12 April. More sVLFEs occurred after the $M_w$ 7.0 Kumamoto earthquake on 16 April, probably triggered by seismic waves of this earthquake, and continued for about a day. The front of the sVLFE activities migrated updip about 4 km per day between 3 and 9 April (Fig. 5a and Supplementary Fig. 4).

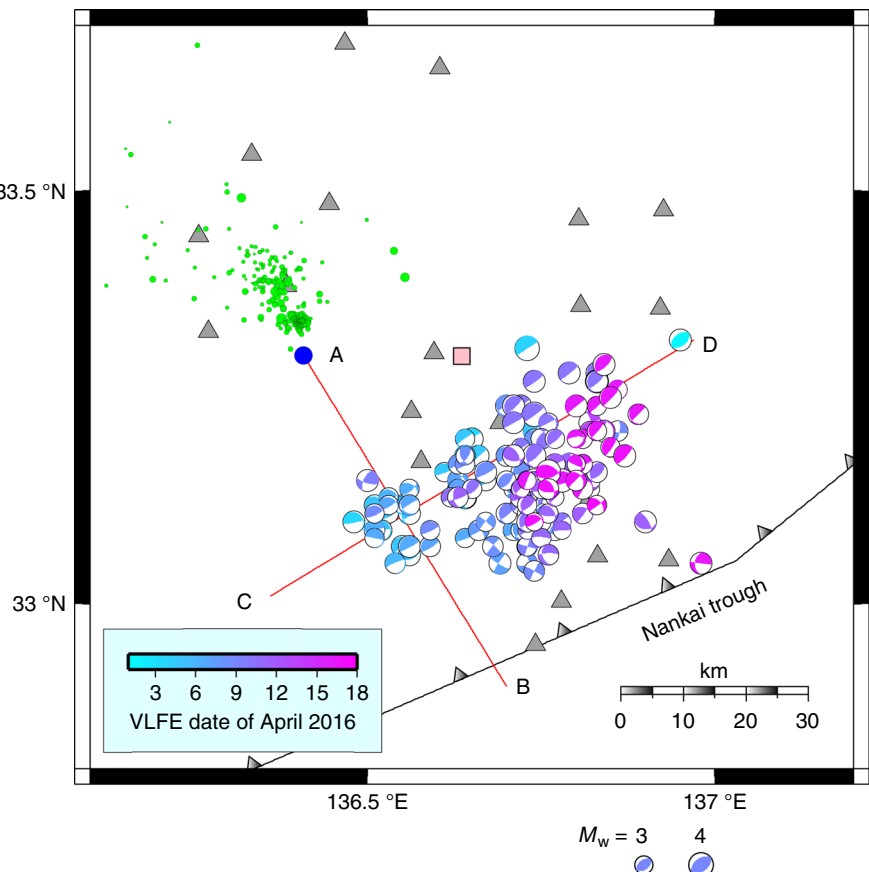

**Fig. 3** Spatiotemporal distribution of sVLFEs in 2016. Focal mechanism symbols represent sVLFE locations; each symbol is colored according to the occurrence date, and the symbol size is proportional to moment magnitude ($M_w$). Blue circle represents the mainshock of the 1 April 2016 off-Mie earthquake; green circles represent aftershocks. Gray triangles indicate DONET stations; pink square is borehole station C0002G. Lines A–B and C–D mark cross sections shown in Fig. 5. Location shown in Fig. 1

This velocity corresponds to a diffusivity of $3 \times 10^3 \, \mathrm{m^2 \, s^{-1}}$ if we assume diffusive migration[44]. The migration velocity is much smaller than that observed for sVLFEs in the Hyuga-nada region[24] or for deep tremor[44]. The sVLFE sources also migrated laterally along the Nankai trough with a similar velocity (Fig. 5b).

Concurrent with the sVLFE activity, transient changes in formation pore pressure in the sedimentary wedge were observed (Fig. 4) at a subseafloor borehole station close to the sVLFE source region (station C0002G; location in Fig. 1 and Supplementary Fig. 1). These pressure changes are consistent with the slip of a shallow SSE updip from station C0002G along the plate boundary[33] (see Methods for descriptions of the observation and SSE estimations), corresponding to the source region of the sVLFEs. Although dehydration of hydrous minerals that increases pore pressure along the plate interface is considered to be one of the causes triggering slow earthquakes[45], the observatory was several kilometers shallower than the plate interface and would not have been affected by such a pressure change. Rather, the observed pore-pressure changes reflect volumetric strain caused by the slip of the SSE, and we assumed that the changes were proportional to SSE moment release (Methods).

The cumulative numbers of sVLFEs and aftershocks of the off-Mie earthquake are compared with the pore-pressure changes in Fig. 4a. The pattern of borehole pore-pressure changes corresponds closely to the cumulative number of aftershocks before 3 April and that of sVLFEs after 3 April, although the trends of the pressure changes are opposite, which point will be discussed later. Even more evident is the similarity of pore-pressure changes and

the cumulative moment of sVLFEs (Fig. 4b). This similarity is not sensitive to the lower limit of magnitude accounted for; thus, it is not affected by the detection limit of small events. The total moment of sVLFEs was $2.9 \times 10^{16}$ Nm for $M_w \geq 3.0$ events, and the estimated moment release due to shallow SSEs was similar order of magnitude[33], between $4 \times 10^{16}$ and $3.2 \times 10^{17}$ Nm. However, we should consider the contributions from smaller sVLFEs, although the size distribution of sVLFEs is not well known (Methods and Supplementary Fig. 5). If sVLFEs follow the Gutenberg–Richter law of regular earthquakes, small events should greatly outnumber large events and the total moment of sVLFEs should be several orders of magnitude larger than the estimate for $M_w \geq 3.0$ events. If instead we assume an exponential distribution of event numbers against seismic moment, as proposed for deep tremor[46, 47], then contributions from smaller events are very limited, as inferred from the curve of the size distribution (Supplementary Fig. 5). For the case of deep slow earthquakes, VLFEs release as much as about 10% of the seismic moment of SSEs[19]. It is difficult to estimate this percentage accurately for shallow events because of the ambiguities in cumulative moment estimations for both sVLFE and SSEs. But the coincidence of sVLFEs and pore-pressure temporal changes and inferred source locations, together with the thrust-type focal mechanisms of sVLFEs, indicate that sVLFEs and pore-pressure changes were driven by SSEs on the same fault rupture along the plate boundary, rather than sVLFEs being triggered at the leading edge of the SSEs. Thus, sVLFEs may represent relatively high-frequency fluctuations of or abrupt stepwise changes in the long-lasting SSE slips.

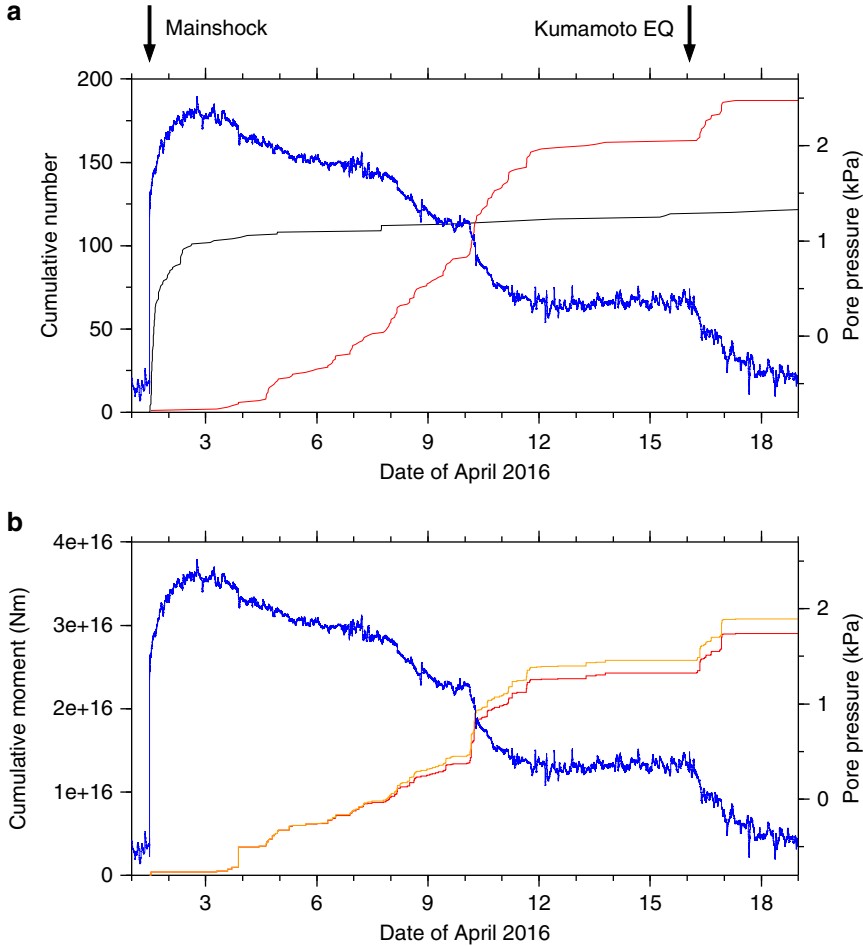

**Fig. 4** Temporal evolution of seismic activities and borehole pore-pressure changes. **a** Cumulative numbers of sVLFEs larger than $M_w$ 3 (red), aftershocks of the off-Mie earthquake larger than magnitude 1 (black), and borehole pore-pressure changes[33] (blue). **b** Temporal evolution of cumulative moment due to sVLFEs larger than magnitude 3.0 (red) and all sVLFEs for which a CMT was determined (orange)

Before 3 April 2016, borehole pore-pressure increased closely corresponding to the cumulative number of aftershocks, which implies that SSEs were occurring at greater depths, around the aftershock region[38]. The subsequent pore-pressure decrease that closely corresponded to the sVLFE activity also supports this idea in suggesting that the slip occurred downdip of the borehole station (Methods). Therefore, if we assume that sVLFEs and SSEs represent a common fault slip, the SSE source migrated updip into the source region of the sVLFEs, then continued updip along with the migration of sVLFE sources. Because previous studies have documented a downdip limit of the sVLFE source in this region[9, 10, 34] (Fig. 1), these relatively deep SSEs were not associated with sVLFEs, or they had almost constant slip rates, with no fluctuations that are detectable as sVLFEs. After 3 April, the source of SSEs became shallower and they were accompanied by sVLFEs. The lateral migration of sVLFE sources along the Nankai trough implies that the SSE source migrated similarly. After 17 April, SSEs continued, but very few sVLFEs were observed. In our interpretation, sVLFE sources also have an updip limit, which SSEs reached at that time.

**Shallow SSEs with or without sVLFEs.** The previous sVLFE episode in 2015 was also accompanied by a pore-pressure changes indicative of shallow SSE. We determined CMTs of these events (Methods, Fig. 1), and the cumulative moment of sVLFEs and borehole pore-pressure changes are compared in Supplementary Fig. 6. The SSE started about 10 days before the sVLFE activity.

Borehole pore-pressure data from two stations available at that time show that the SSE started at a relatively deep location and migrated updip to a shallower depth[33]. As was the case in the 2016 episode, the 2015 SSE was not initially accompanied by sVLFEs. In our interpretation, the upward migration of SSE fault slip reached the sVLFE source region around 21 October 2015, at which time sVLFE activity started and continued for about a week, similar to the 2016 sequence. These observations imply that detectable sVLFEs occurred only when shallow SSEs reached their source region.

Using borehole pore pressure as a proxy for the amount of fault slip occurring during the SSE, we compared daily sVLFE moment releases and the corresponding daily pore-pressure changes (Supplementary Fig. 7). These values show roughly linear correlation with similar values of the proportionality constant for the 2015 and 2016 episodes, with respective cross-correlation coefficients of −0.77 and −0.28. This exercise suggests that the efficiency of sVLFE excitation was similar for these two SSEs, although the ratio on a given day would depend on the detailed slip distribution of a SSE. Although the 2016 SSE was triggered by the off-Mie earthquake or its afterslip, the 2015 SSE did not have a triggering event. This finding suggests that the efficiency of sVLFE excitation does not depend on the triggering events of sVLFEs and SSEs.

**Discussion**
Araki et al.[33] identified eight SSEs in this region during the period 2011–2016 from borehole pore-pressure changes. Among these,

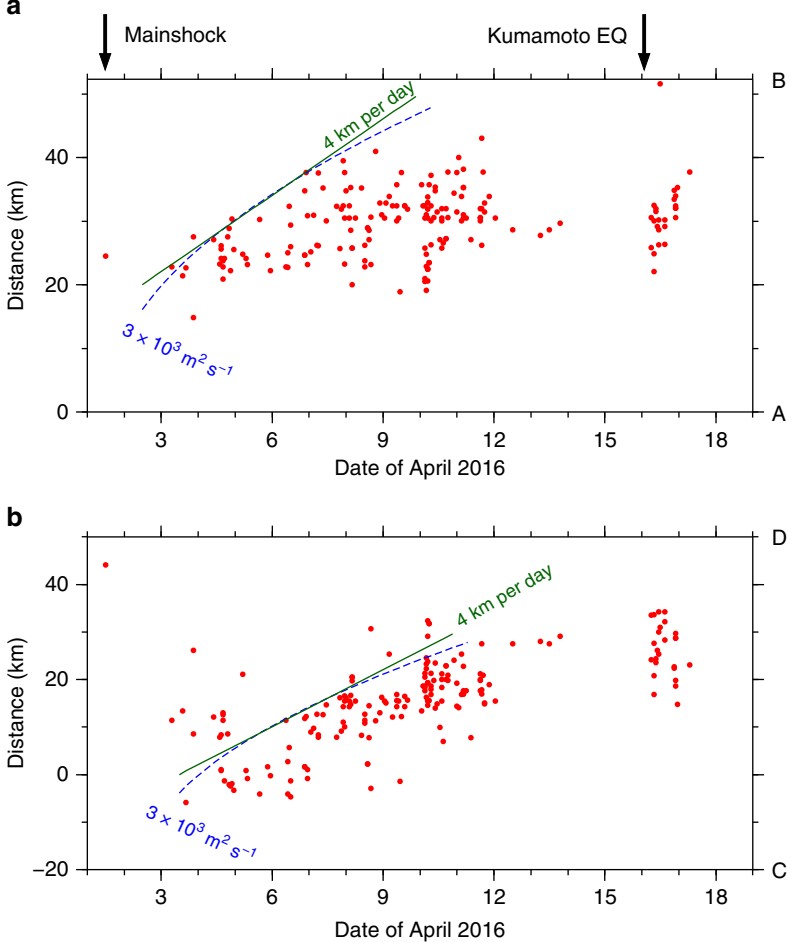

**Fig. 5** Migration of sVLFE source location. **a** sVLFE source locations projected onto line A–B (Fig. 3). Green line indicates migration velocity of 4 km per day; blue curve corresponds to a diffusivity of $3 \times 10^3 \, \mathrm{m^2 \, s^{-1}}$ assuming diffusive migration[44]. **b** sVLFE source locations projected onto line C–D (Fig. 3)

the episodes in 2011, 2015, and April 2016 were estimated to have relatively large slip of 2–4 cm at shallower locations (updip edge of the fault < 24 km landward of the trench), while the other five (two in 2012, two in 2014, and August 2016) had smaller slips of 1–2 cm at deeper locations (updip fault edge >24 km landward of the trench). We checked the DONET continuous seismic records and found that only the three episodes in 2011[37], 2015[10], and April 2016 were accompanied by distinct sVLFE episodes, whereas the five smaller, deeper SSEs were not. Therefore, one condition for the occurrence of distinct sVLFEs is relatively shallow SSE source depth, as observed in the 2015 and 2016 episodes. Another condition may be the magnitude of the SSE, because small SSEs would only excite small sVLFEs, which were not detected. Accordingly, the observation of sVLFEs indicates that the SSE was in a shallow part of the plate boundary; however, the absence of observed sVLFEs does not rule out the occurrence of SSEs, which could be deep or small without triggering observable sVLFEs.

Signals from VLFEs decay with distance $r$ by $r^{-1/2}$ because they are dominantly surface waves, whereas signals from crustal deformation due to SSEs typically decay by $r^{-3}$. Therefore, it remains challenging to detect SSEs offshore beneath the toe of an accretionary prism. The sVLFE episodes observed along the eastern Nankai trough[5, 9, 10, 26, 34, 37] may have accompanied hidden SSEs in their source region. Hidden SSEs also are inferred in connection with sVLFE activity in the Hyuga-nada region[24, 35]. Shallow SSEs may have occurred along the plate interface during

the sVLFE episodes that have been repeatedly observed along the Nankai trough, but SSEs have not been reported because they are difficult to detect. Activities of several sVLFE clusters off the Kii Peninsula and off Shikoku Island have been correlated with intervals of weeks to months[5, 10]. This evidence suggests that SSEs may occur sequentially along the Nankai trough.

In our study, we assumed that pore-pressure changes are proportional to SSE moment release. Heterogeneous slip distribution of SSEs could complicate this assumption. It is not clear whether SSEs occur only near sVLFE clusters or continuously along the subduction zone between sVLFE clusters. This study establishes a correspondence between sVLFEs and shallow SSEs at a single site. Further observations of the full range of slow earthquakes are needed at other sites to investigate physical properties that may be common to different types of slow earthquakes. Finding similarities and differences between deep and shallow slow earthquakes will also improve our understanding of faulting properties along subduction zones. Detailed studies of the distribution and frequency of slow events will be needed to reveal the spatial-temporal coupling along plate boundaries, improve our knowledge of fault slip characteristics, and possibly enable us to monitor the preparatory process of megathrust earthquakes.

## Methods

**Determination of sVLFE sources**. We adopted a CMT waveform inversion method to determine the source location, fault orientation parameters, and

moment magnitude of sVLFEs. We used the SWIFT system[48], which performs the waveform inversion in the frequency domain. A pure double-couple source mechanism is assumed to stabilize the solution. By fitting synthetic waveforms to the observed waveforms for each frequency component, we obtain the best-fitting fault orientation parameters and the location of the source centroid. The normalized residual $E$ between observed and synthesized seismograms is given by

$$E = \frac{\sum_{i=1}^{N_t} \sum_{k=1}^{N_f} w_i \left| \tilde{u}_i^{\mathrm{obs}}(\omega_k) - \tilde{u}_i^{\mathrm{syn}}(\omega_k) \right|^2}{\sum_{i=1}^{N_t} \sum_{k=1}^{N_f} \left| \tilde{u}_i^{\mathrm{obs}}(\omega_k) \right|^2} \quad (1)$$

where $\omega_k$ is the angular frequency, $\tilde{u}_i^{\mathrm{obs}}(\omega_k)$ and $\tilde{u}_i^{\mathrm{syn}}(\omega_k)$ are the Fourier transforms of the observed and synthetic displacement seismograms of the $i$-th trace, $w_i$ is the weighting factor of the $i$-th trace, and $N_t$ and $N_f$ are the numbers of waveform traces and frequency components used for the inversion, respectively. The synthetic waveforms and source-time functions are obtained by an inverse Fourier transform of the estimated parameters in the frequency domain. The moment magnitude is obtained by fitting step-like functions to the obtained source-time function in the frequency range used for the analysis.

We determined CMTs of sVLFEs from the waveforms of ground motion obtained from broadband seismometers (CMG-3TEBB) at each DONET station. Because sVLFE waveforms represent signals that are evident over tens of seconds and also accompany tremor at frequencies of several hertz (Fig. 2), we detected their signals by visually inspecting band-pass filtered waveforms from continuous records. Where sVLFE signals were detected, the waveforms were band-pass filtered between 0.03 and 0.05 Hz, then decimated to a sampling interval of 2 s. We used a total data length of 256 s (128 data points in each channel) for the inversion. The best-fitting source location was obtained by a grid search with horizontal spacing of 0.01° in latitude and longitude, and vertical (depth) spacing of 1 km. We first determined the source centroid location by using velocity seismograms, then estimated the moment tensor at that location by using displacement seismograms. This two-step approach reduced unstable solutions for smaller events due to low signal-to-noise ratios at longer period components. Supplementary Fig. 2 shows an example of CMT inversion and displacement waveform fitting for an event on 10 April 2016. We also determined CMT solutions for the sVLFEs observed in 2015 (Fig. 1).

We also checked waveforms from other DONET observation periods from 2011 to 2016, but we detected no clear sVLFE signals. The single exception was during the activity in 2011[37], for which a CMT could not be determined because an insufficient number of stations recorded the sVLFE signals.

We used the bootstrap method[49] to estimate the standard errors of source locations and focal mechanisms. By using 500 bootstrap replications, in which waveform traces were resampled at random and duplications were not discarded, we estimated the errors of the CMT inversion (Supplementary Fig. 8). The average source location errors were 5 km in the horizontal and 2 km in the vertical direction, although errors were larger for events that were deeper or outside the observation network. The average errors of the plunge and azimuth of the principal axes were 16° and 29°, respectively, regardless of event depths.

The magnitude–frequency distribution of the April 2016 sVLFEs is shown in Supplementary Fig. 5, one plot assuming the power law (the Gutenberg–Richter law) for regular earthquakes and the other assuming the exponential distribution obtained for deep tremor[46, 47]. Neither distribution fits the data well, although statistical fits cannot be calculated given the small number of events. The $b$-value (negative of the slope of the power-law distribution) for this activity is about 2, much larger than the typical value of about 1 for regular earthquakes. Previous studies have found similar size distributions of shallow or deep VLFEs[27, 50].

**Borehole pore-pressure changes and SSEs in the Nankai trough**. Two subseafloor borehole observatories (C0002G and C0010A) are installed in the sedimentary wedge above the plate interface in the Nankai trough off the Kii Peninsula[33, 51, 52]. They continuously monitor formation pore pressure in fine-grained and low-permeability sediments. Pore-fluid pressure sensors are installed in two hydraulically isolated depth intervals (757–780 and 931–980 m below seafloor; mbsf) at station C0002G (1937.5 m seafloor depth), and in the 389–407 mbsf interval at station C0010A (2523.7 m seafloor depth). Pore-pressure data have documented repeated SSEs along the Nankai trough[33]. In the following, we briefly summarize the method used to estimate SSE slip from observed pore-pressure changes[33].

The factors to be considered as causes of pore-pressure changes include coseismic and environmental effects. One factor in coseismic change is the movement of underground water due to nonlinear mechanical behavior of sediments or aquifer channels under strong earthquake shaking[53]. Decreases of effective normal stress across fault interfaces due to water intrusion could cause triggered seismicity. This effect is unlikely in our data because the observatories are in consolidated sedimentary rocks without connections to aquifer channels, and most previously reported pressure transients[33] did not accompany strong shaking due to nearby large earthquakes. The other coseismic factor to be considered is local volumetric strain due to fault slip[54]. Because borehole pore pressures are measured in the accretionary prism several kilometers above the plate interface, the pore-pressure data should indicate strain changes due to slip along the plate boundary. Pressure changes due to ocean tides and other oceanographic loading

were removed by the use of a colocated reference pressure sensor on the seafloor. Accordingly, we accept pore-pressure transients as reflecting volumetric strain caused by slip on the plate interface.

The pressure-to-strain conversion factor is 5.7 kPa per microstrain at C0002G and 4.7 kPa per microstrain at C0010A. Using data from two borehole sites, Araki et al.[33] obtained the following SSE fault models on the basis of forward modeling rather than data inversion. Assuming that SSEs occurred on the plate boundary, 1–4 cm of slip on a patch 20–40 km wide consistently explains the pressure transients at the two sites for the episodes observed from 2011 to 2016. The polarity of volumetric strain, and accordingly the pore-pressure change, is reversed updip and downdip on the fault. Araki et al.[33] used this information to locate the SSE fault patch and trace its migration during the 2015 SSE episode. For the SSE episode after the 2016 off-Mie earthquake, they obtained a solution with 2–4 cm slip updip of station C0002G, corresponding to a total moment release between $4 \times 10^{16}$ and $3.2 \times 10^{17}$ Nm ($M_w$ 5.0–5.6).

This methodology allows us to assume a proportionality between pore-pressure changes and fault slip, and hence the moment release during SSE episodes on a fault with a fixed location. A migrating SSE slip patch would complicate this relation, because the amount of volumetric strain depends on the direction and distance from the fault. By using data from site C0002G located downdip from the sVLFE source area, we could assume a proportionality between pore-pressure changes and SSE moment release as a first-order approximation. Data from station C0010A, located above the sVLFE source area, were not obtained during the 2016 SSE episode[33].

**Data availability**. DONET and borehole observation data are publicly available from https://join-web.jamstec.go.jp/join-portal/en/. DONET data are also available from http://www.hinet.bosai.go.jp/?LANG = en.

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

## Acknowledgements

We appreciate the efforts of the people responsible for the development and maintenance of DONET. We also thank Drs. H. Sugioka and Y. Ito for sharing the source parameters of sVLFE. This work was partly supported by MEXT KAKENHI (16H06477). All figures were drawn using Generic Mapping Tools (GMT)[55].

## Author contributions

M.N. designed the study, carried out the data analysis, proposed the model, and wrote the paper with contributions from all coauthors. T.H., E.A., S.K., and S.I. organized the geophysical interpretations. All authors contributed equally to scientific discussions.

## Additional information

**Competing interests:** The authors declare no competing interests.

