## [Peer Review File · Nature Communications]

Reviewers' comments:

Reviewer #1 (Remarks to the Author):

The manuscript presents shallow VLFs in Nankai trough using OBSs, and correlates their activity with pore pressure changes to argue that SSE and VLFs are occurring at the same fault and products of the same process. This study involves one of the very interesting and critical aspects of fault slip – slow earthquakes – and presents observations that are very challenging to make. However, there are some improvements to be made to support main conclusions of this manuscript.

I recommend publication after addressing the issues described below.

Major issue:

The only evidence presented for SSE is pore pressure change data from one station. It does appear to show changes that correlates with VLFE activities in time, but not always, for example first few days of fig 3, S5, S7, and S8. Even if the temporal correlation is real, the spatial correlation is essentially an assumption. Authors argue that slow slip migrates from downdip to updip, and in the process, activates VLFE in certain areas only. Unfortunately, slow slip may not be the only reason for pressure changes. Authors should consider other possibilities and discuss them. Even if the pressure change is related to slow slip, authors should show that the pattern of pressure change is consistent with migrating slip on the same fault plane, the plate interface, I presume. In fact, there are significant heterogeneity in the focal mechanism indicating involvement of multiple faults. I suggest modeling of pore pressure change due to slow slip to provide more convincing evidence to support the conclusion.

Minor issues:

Fig 1: Use different colors for focal mechanisms instead of different shades of grey

In the introduction, importance of shallow slow earthquakes should be discussed in the context of fault physics and earthquake/tsunami hazards

Line 48-50: authors should note the models for VLFs proposed by Gombert et al., 2016 and occurrence of asynchronous deep VLFE and tremor by Hutchison and Ghosh, 2017. They may be inconsistent to the model suggested here, but it does not weaken the findings of this manuscript in any way. Presenting a complete picture is important to evaluate the current results.

I am wondering if the authors tried to find tremor and/or LFE activity during the time periods in question. They may provide additional constraints on the spatial correlation.

Reviewer #2 (Remarks to the Author):

This is an interesting paper. My primary concern about the presentation is that the authors have not clearly delineated what is inference from what is observation. The novel observations are of temporal correlations between pore pressure changes in a single borehole and the rates of numbers or moment release of sVLFs located nearby. These are both inferred to be caused by a SSE.

However, as written, it is often not clear that the SSE behavior and its relationship to BOTH the pore pressures and sVLFs are inferences. In my opinion, the paper should be posed as a test of the hypothesis that BOTH the pore pressures and sVLFs may be a response to a single SSE; instead the SSE is assumed and hypothesized to cause the sVLF activity. There is a single statement that the pore pressures are a proxy for an SSE, without any justification provided (e.g. why does the SSE have to be beneath the borehole, could some other process cause the pore

pressures?). This needs to be remedied, and the authors need to demonstrate that a record of pressure changes at a single site constrain the occurrence of a SSE directly beneath it.

Additionally, the presentation could lead the reader to believe that the authors have set out to demonstrate that their inference of SSEs causing sVLFs is correct, which leads only to a weak conclusion of plausibility. The conclusions would be much stronger if a hypothesis was clearly stated (i.e. that noted above), and tests described that attempted to reject it. In other words, what else might explain the observations of pore pressure changes and sVLFs? Could the temporal correlations shown just be chance? How unique is their model?

I have annotated the manuscript to try to make the manuscript reflect the above concerns and with other suggestions and make additional comments below.

Several times in the introductory paragraphs the authors state that sVLFs have been poorly studied because they've rarely been observed. This seems to contradict the sentence on line 68, stating "Along the Nankai trough, sVLFs have been repeatedly detected" with a list of eight references!

The discussion of the cumulative numbers and moments of the estimated sVLFs in lines 106-124 would benefit from some rewriting and shortening. It seems sufficient simply to show these two quantities for all the events and only those for $M > 3.0$. If stated that magnitude is proportional to the log of the moment, then it will be clear why the small events have a much smaller contribution to the cumulative moment.

Lines 106-129 discuss the fraction of moment in the SSE released by sVLFs. However, nothing presented to this point shows or discusses how moment of the SSEs was estimated? The only reference to a measurement of an SSE (aseismic slow slip) is of the volumetric strain recorded at a single site, at borehole station C0002G. Lines 124-129 cite percentages of SSE moment released as sVLFs – where does the moment of the SSE come from?

Why does the Kumamoto earthquake coincide with a sharp decline in the pore pressures when the off-Mie earthquake coincides with a sharp increase?

The authors state that one condition for sVLFs is the concurrent occurrence of an SSE beneath the sVLF source region, and further that there is an approximate scaling between the size of the SSE and size of the sVLF (lines 173-181). This is only true if sVLFs are only observed when SSEs are also observed, yet the authors do not describe how sVLFs were detected, so this cannot be assessed. Later in the text, the authors state that there are SSEs without sVLFs but are there sVLFs without indications of SSEs (e.g. with no pressure changes)? Lines 187-190 suggest that this is the case, but it is unclear whether this was the case in this particular study. To be able to evaluate whether the sVLFs rates are really likely responses to SSEs (or inferred SSEs) it is important to describe how sVLFs were searched for, which was not done in

The authors infer that an sVLF can only be generated by a collocated SSE, otherwise it is not possible to state that the size of the sVLF scales with the size of the SSE (lines 177-179). This needs to be better justified. First, it is not clear if it is the size (moment?) of an individual sVLF signal or the rate of sVLFs that do or should scale with the SSE's attributes. Second, does the 'size' of the SSE refer to its moment, its area, slip, etc?. Finally, why do they need to be collocated, given that stress transfer and secondary processes (e.g. pore pressure migration) can occur over some distances.

Reviewer #3 (Remarks to the Author):

The manuscript "Shallow very-low-frequency earthquakes accompany slow slip events, Nankai trough, Japan" by Nakano-san et al documents an interesting set of observations of very low

frequency earthquakes along the Nankai trough detected by the DONET network. Due to the seafloor network, the authors are able to constrain focal mechanism for a sequence of shallow VLFs and compare their migration and moment release with pore pressure changes suggesting the occurrence of a SSE at the same time. They suggest that the VLFs represent high frequency fluctuations in slip and can therefore be used to identify periods of shallow SSE activity which may not be detectable by other methods. I enjoyed reading the manuscript which is generally well written. The analysis conducted by the authors seems robust and if quantifiable the observations could be used as a proxy for detecting shallow SSEs which may not be detectable geodetically. While it does seem that there is a correlation between the VLFs and SSE (based on the pore pressure changes) I would like to see some additional analysis to firm up the authors conclusions and perhaps discuss what the implications may be on stressing rates for triggering.

Detailed Comments:

- Could the authors comment on the cluster of events which appear to show more of a strike slip mechanism? There seems to be similar mechanisms from the 2015 event in the same location. Could these be related to the quite strong along strike migration?
- You mention the migration rate in the updip direction, how does it compare with the along strike rate? Does this have implications on the overall slip direction of the SSE if they're related?
- I agree that the similarity between cumulative moment is probably clearer than just the cumulative number of events. I would suggest moving/merging figure S5 into the main article. The drop in pore pressure and increase in moment release on 10th April is quite striking. Was there a larger event here with its own set of aftershocks?
- At the moment the comparison between moment release and pore pressure change is quite qualitative. To give more certainty on the results, could the authors do some statistical analysis on the correlation between the two datasets? This could also be done on the 2015 events.
- Do you envisage that the VLFs are the result of stress triggering at the leading edge of the SSE as it migrates?
- If the VLFs are occurring along same fault plane and only begin when the SSE gets close to their source region, are you able to make some inferences about the stressing rates required to trigger the VLFs? Could you make an estimate on the slip size for the 2011, 2015 and 2016 events and the potential stressing rates vs the early events which didn't have accompanying VLFs? The static stress increase around the SSE is probably not insignificant but as it rapidly decays with distance you may be able to infer something about how close the SSE needs to get to the source region for the VLFs to occur?

Minor comments

Could you make the pore pressure gauge larger in figure 1 or give it a label? It was hard to find.

Could the authors try to give some indication on the variation in magnitude of the VLFs in figure 2? It might be useful to see whether there was a change in magnitude as the sequence migrated.

As I mention above, consider merging or at least adding the cumulative moment plot into figure 2 from figure S5.

Given the large variations in pressure how confident are you in the comparison between the ratio in the daily changes? Can you attach some uncertainty to the values?

Responses to the comments from reviewers.

Reviewers' comments:

Reviewer #1 (Remarks to the Author):

The manuscript presents shallow VLFES in Nankai trough using OBSs, and correlates their activity with pore pressure changes to argue that SSE and VLFES are occurring at the same fault and products of the same process. This study involves one of the very interesting and critical aspects of fault slip – slow earthquakes – and presents observations that are very challenging to make. However, there are some improvements to be made to support main conclusions of this manuscript.

I recommend publication after addressing the issues described below.

Major issue:

The only evidence presented for SSE is pore pressure change data from one station. It does appear to show changes that correlates with VLFES activities in time, but not always, for example first few days of fig 3, S5, S7, and S8. Even if the temporal correlation is real, the spatial correlation is essentially an assumption. Authors argue that slow slip migrates from downdip to updip, and in the process, activates VLFES in certain areas only. Unfortunately, slow slip may not be the only reason for pressure changes. Authors should consider other possibilities and discuss them. Even if the pressure change is related to slow slip, authors should show that the pattern of pressure change is consistent with migrating slip on the same fault plane, the plate interface, I presume. In fact, there are significant heterogeneity in the focal mechanism indicating involvement of multiple faults. I suggest modeling of pore pressure change due to slow slip to provide more convincing evidence to support the conclusion.

Responses: We have revised the manuscript following the reviewer's comments. First, we have added brief summaries of the borehole pore-pressure observations and their interpretations in Methods. This includes the methods to estimate the location and magnitude of SSEs from the pore-pressure changes, which is after Araki et al. (2017). Using the polarity of pore-pressure changes, location of SSE slip can be obtained (L. 324). The initial increase and following decrease of pore-pressure changes in the 2016 sequence represent that the slip of SSEs occurred downdip first then moved to updip of the borehole station C0002G. Because of this observation, we consider that SSEs fault have migrated from downdip to updip along the plate interface. SSEs downdip around

the aftershock region of the off-Mie earthquake was also pointed out by Wallace et al. (2016). The SSEs slip updip of C0002G overlaps the observed sVLFE source region and we can assume the spatial correlation of these slow events.

The absence of sVLFE activities in the initial stage of SSEs have been interpreted that sVLFE was not triggered during the deeper SSEs because sVLFE is considered to have downdip limit of the activity as also observed in the previous studies. This point has already been discussed in the original manuscript (L. 166).

In Methods, we also have discussed other possibilities causing pore-pressure changes as coseismic nonlinear responses of the media or other environmental effects, which are not considered to be applied in our observations (L. 304).

The variation of sVLFE focal mechanisms were mainly found for smaller magnitude events, of which uncertainty of CMT solution was larger because of smaller number of waveforms used for the inversion. The diversity of focal mechanism would also be caused by complexities of plate boundary fault geometry due to subducted seamounts (Park et al. 2003). This point has been added to the revised manuscript (L. 101)

Minor issues:

Fig 1: Use different colors for focal mechanisms instead of different shades of grey

Response: We have revised Fig. 1 according to the reviewer's comment.

In the introduction, importance of shallow slow earthquakes should be discussed in the context of fault physics and earthquake/tsunami hazards

Response: We have added passages describing the importance of studies about shallow slow earthquakes in introduction (L. 33-41).

Line 48-50: authors should note the models for VLFs proposed by Gomberg et al., 2016 and occurrence of asynchronous deep VLFs and tremor by Hutchison and Ghosh, 2017. They may be inconsistent to the model suggested here, but it does not weaken the findings of this manuscript in any way. Presenting a complete picture is important to evaluate the current results.

Response: We have added several references related to slow earthquake observations other than Nankai trough and different source models (L. 54-56 and 60-64).

I am wondering if the authors tried to find tremor and/or LFE activity during the time periods in question. They may provide additional constraints on the spatial correlation.

Response: The observed sVLFs accompany tremor signals as shown in Fig. 2. Since the tremor signals related to SSEs have already been studied by Araki et al. (2017), we have not analyzed them.

Reviewer #2 (Remarks to the Author):

This is an interesting paper. My primary concern about the presentation is that the authors have not clearly delineated what is inference from what is observation. The novel observations are of temporal correlations between pore pressure changes in a single borehole and the rates of numbers or moment release of sVLFs located nearby. These are both inferred to be caused by a SSE.

Response: According to the reviewer's comment, we have revised the manuscript throughout to clarify the observations and assumptions in our study. The observations are CMT solutions of sVLFs, time series of their cumulative moment, and borehole pore-pressure changes, while the assumptions are the occurrence of SSEs and the proportionality of their moment release to borehole pore-pressure changes. Actually, SSEs have already been estimated by Araki et al. (2017), which point have been added in Methods (L. 294-).

However, as written, it is often not clear that the SSE behavior and its relationship to BOTH the pore pressures and sVLFs are inferences. In my opinion, the paper should be posed as a test of the hypothesis that BOTH the pore pressures and sVLFs may be a response to a single SSE; instead the SSE is assumed and hypothesized to cause the sVLF activity. There is a single statement that the pore pressures are a proxy for an SSE, without any justification provided (e.g. why does the SSE have to be beneath the borehole, could some other process cause the pore pressures?). This needs to be remedied, and the authors need to demonstrate that a record of pressure changes at a single site constrain the occurrence of a SSE directly beneath it.

Response: To answer these questions, we have added brief summaries of the borehole pore-pressure observations and interpretations of the data in Methods, which is based on Araki et al. (2017). The observed pore-pressure changes are caused by volumetric strain due to SSE slip along the plate interface, and can be assumed to be proportional to its moment release for the first order approximation (L. 334). Using the polarity of pore-pressure changes, location of SSE slip can be obtained (L. 324). The initial increase and following decrease of pore-pressure changes in the 2016 sequence represent that the slip of SSEs occurred downdip first then moved to updip of the borehole station

C0002G. The SSE slip updip of C0002G overlaps the observed sVLF E source region and we could assume the spatial correlation of these slow events.

Based on these considerations, the observed coincidence of temporal changes of the pore-pressure and the sVLF E cumulative moment, and thrust-type focal mechanisms of sVLF Es, we concluded that sVLF Es and SSEs shared the same fault slip along the plate boundary.

Additionally, the presentation could lead the reader to believe that the authors have set out to demonstrate that their inference of SSEs causing sVLF Es is correct, which leads only to a weak conclusion of plausibility. The conclusions would be much stronger if a hypothesis was clearly stated (i.e. that noted above), and tests described that attempted to reject it. In other words, what else might explain the observations of pore pressure changes and sVLF Es? Could the temporal correlations shown just be chance? How unique is their model?

Responses: According to the reviewer's comment, we have discussed other factors that could cause pore-pressure changes in Methods (L. 304), which are not considered to be plausible in our observations. We have computed the cross-correlation coefficient of the daily sVLF E cumulative moment and daily pore-pressure changes (L. 189), which shows statistical significance of the correlations of sVLF E moment release and pore-pressure changes (and accordingly SSE moment release), especially in the 2016 episode.

I have annotated the manuscript to try to make the manuscript reflect the above concerns and with other suggestions and make additional comments below.

L. 48. Didn't make sense to say that 'slow earthquakes are accompanied by other types of slow earthquakes'.

Response: As shown in references cited following this sentence, many observations have shown that occurrences of different types of slow events are spatially and temporally correlated each other. We call these phenomena 'accompany'.

L. 52. Unclear what the distinction is between sVLF Es and SSEs are in these two sentences? Please clarify.

Response: The definitions and distinction of sVLF Es and SSEs have already given in the text.

L. 67. Please clarify which activity you're referring to.

Response: This passage considers sVLFE activity. We have revised the manuscript accordingly (L. 78).

L. 67. It is unclear if these are results from other studies, or this one. If the former, please add references.

Response: This passage refers to results from other studies. We have added citations of related studies (L. 82).

L. 72. Is an 'episode' a period of 20 minutes with 5 events? To me the term implies a longer, more significant event. Maybe 'short burst' would be more appropriate?

Response: Shown in this figure is an example of a burst activity during the episode. We have corrected the manuscript to clarify this point (L. 88).

L. 75. Could the timing relative to this earthquake be a coincidence?

Response: We still do not have enough data or source models to identify whether the off-Mie earthquake trigger the SSE or these events are just a coincidence. However, we consider it is reasonable to adopt the former interpretation because we have several observations that significant sVLFE activities preceded by shakings from close or distant large earthquakes in 2004 (Obara and Ito, 2005), 2011 (To et al. 2015), and 2016 (Fig. 4). We have revised the manuscript concerning a possibility of the latter case (L. 83).

L. 92. How well and accurately are the sVLFs located?

Response: To estimate the accuracy of the source location, we have carried out the bootstrap error estimations, which point has been added in Methods (L. 279). We also have added a figure (Supplementary Fig. 4) showing the error ellipsoids of source locations. The errors were 5 km in horizontal direction and 2 km in vertical direction, for averages of all the events shown.

L. 124. It was just stated that deep studies show a maximum of about 10%, which doesn't seem "large" to me!

Response: I agree to the reviewer's comment. Since we have not obtained the ratio, we have revised the related passages to only mention that sVLFE and SSE are considered to have common fault slip (L. 153-156).

L. 145. Can barely see the sVLFE moment curve in the figure S7.

Response: We have revised the figure to clearly show the sVLFE moment curve (Supplementary Fig. 7).

L. 152. Is there independent evidence of SSE migration?

Response: The migration of the SSEs in 2015 has been identified from the polarity change of one of two borehole pore-pressure changes by Araki et al. (2017). In 2016, Wallace et al. (2016) inferred the occurrence of SSEs around the aftershock region of the off-Mie earthquake. The initial increase and following decrease of pore-pressure changes also consistently represent that the slip of SSEs occurred downdip first then migrated to updip of the borehole station C0002G. These results are obtained independent of the sVLFE source migration.

L. 166. Please state what the evidence for this is, so the reader does not have to look up and read the referenced study. It is important to know that the SSE location and migration is determined independently of the sVLFs, to be able to conclude that the inferred linkages are causal.

Response: We have added a brief summary of location, depth, and slip of SSE for these events estimated by Araki et al. (2017) (L. 199-204).

L. 178. Were sVLFs observed without independent evidence of SSEs?

Response: This passage describes possibilities of hidden SSEs during past sVLFE activities. We have added references for the past sVLFE observations (L. 218).

Several times in the introductory paragraphs the authors state that sVLFs have been poorly studied because they've rarely been observed. This seems to contradict the sentence on line 68, stating 'Along the Nankai trough, sVLFs have been repeatedly detected' with a list of eight references!

Response: Five of eight references were based on land observations to locate sVLFE events. These studies offered poor resolutions for offshore events.

The discussion of the cumulative numbers and moments of the estimated sVLFs in lines 106-124 would benefit from some rewriting and shortening. It seems sufficient simply to show these two quantities for all the events and only those for $M > 3.0$. If stated that magnitude is proportional to the log of the moment, then it will be clear why

the small events have a much smaller contribution to the cumulative moment.

Response: We have revised the figure according to the reviewer's comment (newly Fig. 4).

Lines 106-129 discuss the fraction of moment in the SSE released by sVLFs. However, nothing presented to this point shows or discusses how moment of the SSEs was estimated? The only reference to a measurement of an SSE (aseismic slow slip) is of the volumetric strain recorded at a single site, at borehole station C0002G. Lines 124-129 cite percentages of SSE moment released as sVLFs; where does the moment of the SSE come from?

Response: The observed pore-pressure changes can be converted to volumetric strain due to SSE fault slip. Assuming size of the SSE fault, the seismic moment of SSE can be obtained. We have added a summary of their estimations in method section (L. 327-329).

Why does the Kumamoto earthquake coincide with a sharp decline in the pore pressures when the off-Mie earthquake coincides with a sharp increase?

Response: The polarity of pore-pressure changes depends on the location of fault slip relative to the observatory (see L. 324). The coseismic increase in pore-pressure due to the off-Mie earthquake is interpreted to SSEs occurred around the deeper aftershock region as pointed out by Wallace et al. (2016) (L. 160). On the other hand, the decrease of the pore pressure after the Kumamoto earthquake is considered as accelerations of the ongoing SSEs occurring in the shallower part.

The authors state that one condition for sVLFs is the concurrent occurrence of an SSE beneath the sVLF source region, and further that there is an approximate scaling between the size of the SSE and size of the sVLF (lines 173-181). This is only true if sVLFs are only observed when SSEs are also observed, yet the authors do not describe how sVLFs were detected, so this cannot be assessed. Later in the text, the authors state that there are SSEs without sVLFs but are there sVLFs without indications of SSEs (e.g. with no pressure changes)? Lines 187-190 suggest that this is the case, but it is unclear whether this was the case in this particular study. To be able to evaluate whether the sVLFs rates are really likely responses to SSEs (or inferred SSEs) it is important to describe how sVLFs were searched for, which was not done in

Response: We have detected sVLFs by visual inspections of band-passed waveforms. This point has been added in Methods (L. 260-263).

The authors infer that an sVLF can only be generated by a collocated SSE, otherwise it is not possible to state that the size of the sVLF scales with the size of the SSE (lines 177-179). This needs to be better justified. First, it is not clear if it is the size (moment?) of an individual sVLF signal or the rate of sVLFs that do or should scale with the SSE's attributes. Second, does the 'size' of the SSE refer to its moment, its area, slip, etc?. Finally, why do they need to be collocated, given that stress transfer and secondary processes (e.g. pore pressure migration) can occur over some distances.

Responses: Our observations showed that the detected sVLF activities off the Kii Peninsula in 2015 and 2016 accompanied collocated SSEs with very similar temporal changes of the moment release, assuming a proportionality of the pore-pressure changes to SSEs moment release (justification given in L. 330-337). Daily changes of these values were also well correlated (L. 189-191 and Supplementary Fig. 8), although the data points were scattered. These observations, however, would not exclude the possibilities of sVLF without SSE, and we have added references of such observations (Wech and Bartlow, 2014; Hutchison and Ghosh, 2016; L. 63).

As pointed out by the reviewer, decrease of effective normal stress across fault interfaces due to water intrusion could cause triggered seismicity. However, our pore-pressure observations are conducted at sites several kilometers shallower than the plate boundary fault and would not have been affected by such a pressure change. Rather, the pore-pressure change reflect volumetric changes due to fault slip. Therefore, the sVLF activity is not considered as triggered one due to stress transfer or migration of pore fluids. This point has been added in methods section (L. 126-132).

Reviewer #3 (Remarks to the Author):

The manuscript "Shallow very-low-frequency earthquakes accompany slow slip events, Nankai trough, Japan" by Nakano-san et al documents an interesting set of observations of very low frequency earthquakes along the Nankai trough detected by the DONET network. Due to the seafloor network, the authors are able to constrain focal mechanism for a sequence of shallow VLFs and compare their migration and moment release with pore pressure changes suggesting the occurrence of a SSE at the

same time. They suggest that the VLFEs represent high frequency fluctuations in slip and can therefore be used to identify periods of shallow SSE activity which may not be detectable by other methods. I enjoyed reading the manuscript which is generally well written. The analysis conducted by the authors seems robust and if quantifiable the observations could be used as a proxy for detecting shallow SSEs which may not be detectable geodetically. While it does seem that there is a correlation between the VLFEs and SSE (based on the pore pressure changes) I would like to see some additional analysis to firm up the authors conclusions and perhaps discuss what the implications may be on stressing rates for triggering.

Detailed Comments:

• Could the authors comment on the cluster of events which appear to show more of a strike slip mechanism? There seems to be similar mechanisms from the 2015 event in the same location. Could these be related to the quite strong along strike migration?

Response: We have added new figures showing depth distribution and source location errors of sVLFE (Supplementary Figs. 3 and 4). The sVLFEs showing strike-slip mechanism were located deeper than 10 km, but the source location errors were relatively large. The diversity of focal mechanism implies some complexity of fault geometry due to small-scale structural heterogeneities such as subducted seamounts (Park et al. 2003). This point has been added in the revised manuscript (L. 99-104).

• You mention the migration rate in the updip direction, how does it compare with the along strike rate? Does this have implications on the overall slip direction of the SSE if they’re related?

Response: sVLFE sources also seem to show along-strike migrations, of which velocity is similar to the value of updip migration. This would imply that SSE source also have migrated along strike of the subducting plate. We have added this point in the revised manuscript (L. 170).

• I agree that the similarity between cumulative moment is probably clearer than just the cumulative number of events. I would suggest moving/merging figure S5 into the main article. The drop in pore pressure and increase in moment release on 10th April is quite striking. Was there a larger event here with its own set of aftershocks?

Response: According to the reviewer's comment, we have moved the graph of

sVLFs cumulative moment to the figure showing a comparison with the cumulative number (newly Fig. 4). We did not find any distinct activities or larger events in the aftershock activity around 10 April.

• At the moment the comparison between moment release and pore pressure change is quite qualitative. To give more certainty on the results, could the authors do some statistical analysis on the correlation between the two datasets? This could also be done on the 2015 events.

Response: We have computed the cross-correlation coefficients between the daily changes of the moment release and pore pressures. The cross-correlation coefficient is -0.77 and -0.28 for the 2015 and 2016 episodes, respectively. We have added this point in the revised manuscript (L. 189-191).

• Do you envisage that the VLFs are the result of stress triggering at the leading edge of the SSE as it migrates?

Response: Although increased pore pressure along the plate interface due to dehydration of hydrous minerals is considered to be one of the causes triggering slow earthquakes (e.g. Shelly et al. 2006), the observed pressure change would not reflect this because the observatory was located at several kilometers shallower than the plate interface fault. Rather, the observed pore-pressure changes reflect volumetric strain caused by the slip of SSEs, and it is reasonable to consider that sVLFs and SSEs shared the same fault rupture along the plate boundary. We have added these points in the revised manuscript (L. 126-132)

• If the VLFs are occurring along same fault plane and only begin when the SSE gets close to their source region, are you able to make some inferences about the stressing rates required to trigger the VLFs? Could you make an estimate on the slip size for the 2011, 2015 and 2016 events and the potential stressing rates vs the early events which didn't have accompanying VLFs? The static stress increase around the SSE is probably not insignificant but as it rapidly decays with distance you may be able to infer something about how close the SSE needs to get to the source region for the VLFs to occur?

Response: We have fitted a line to the daily changes of the sVLF moment release

and pore-pressure changes, a proxy for slip or moment of SSEs, for 2015 and 2016 sequences (L. 189 and Supplementary Fig. 8). Since these lines pass through around (0, 0), minimum stressing rate necessary to trigger sVLFE would not exist. However, further detailed studies are necessary because large error was found for estimations of the daily pore-pressure changes.

We consider that the absence of sVLFE in the early stage of these activities is because the SSEs occurred at depths deeper than the downdip limit of sVLFE activities. This point has already been discussed in the original manuscript (L. 166-169 and 207).

Minor comments

Could you make the pore pressure gauge larger in figure 1 or give it a label? It was hard to find.

Response: Since Fig. 1 has already included many information, adding station code further complicates the figure. Rather, we have added the borehole station in Supplementary Fig. 1.

Could the authors try to give some indication on the variation in magnitude of the VLFEs in figure 2? It might be useful to see whether there was a change in magnitude as the sequence migrated.

Response: The mechanism symbol has already been proportional to the magnitude. We have added legends to indicate the magnitude.

As I mention above, consider merging or at least adding the cumulative moment plot into figure 2 from figure S5.

Response: We have merged these figures in a new Fig. 4.

Given the large variations in pressure how confident are you in the comparison between the ratio in the daily changes? Can you attach some uncertainty to the values?

Response: We have conventionally computed the errors of daily changes from the standard deviation of pore-pressure changes, which have been added to the figure (Supplementary Fig. 8).

Reviewers' comments:

Reviewer #2 (Remarks to the Author):

The authors have addressed many of the concerns noted by the reviewers. Although a bit clearer now, in many instances the presentation still conveys the idea that the sVLFs and SSEs are the observations, rather than the observations being the sVLFs and pressure changes, and the SSEs being inferred from the pressure changes. I have suggested some modified prose in a few places in the accompanying annotated manuscript, which are intended to make this distinction clearer,,,,,, and hope that the authors would make a final attempt to make it clearer elsewhere as well.

The one concern raised by both myself and another reviewer, was that the authors do not present a serious consideration of other interpretations. They have added text in lines 54-56 and 60-64, but these new lines basically just note that different combinations of types of slow earthquakes occur in different places and times, and thus that "slow earthquakes have complicated source processes". I found Figure 2 striking evidence that at least would be consistent with the VLFs being consistent with the model presented in Gombert et al. (2016), as it is definitely clear from Figure 2 that the VLFs could represent the envelope of clustered LFEs. The diversity of focal mechanisms (see below) further supports this alternative interpretation. Acknowledging alternative models does not negate the one the authors clearly prefer, but a more objective presentation would demonstrate that other models have been considered and perhaps would even say why the one selected was preferred.

A related disturbing section of the paper is the discussion of the focal mechanisms, and fact that the authors present rather weak excuses for simply ignoring the mechanisms that do not agree with their model! The significant diversity of focal mechanisms is attributed to error and heterogeneity, yet some of the mechanisms are extremely different and yet somewhat spatially consistent (e.g. group of strike-slip events in darker blues). This would imply that many of the focal mechanisms are completely wrong, as it is hard to imagine what 'heterogeneity in the plate interface' can explain them! The authors simply "assume that the sVLFs generally represent low-angle thrust movement on the plate interface", despite evidence to the contrary!

Reviewer #3 (Remarks to the Author):

Thank to the authors for their efforts in revising the manuscript. I am happy that all of the comments raised have been suitably addressed in the revised manuscript.

Responses to the comments from reviewers.

Reviewers' comments:

Reviewer #2 (Remarks to the Author):

The authors have addressed many of the concerns noted by the reviewers. Although a bit clearer now, in many instances the presentation still conveys the idea that the sVLFs and SSEs are the observations, rather than the observations being the sVLFs and pressure changes, and the SSEs being inferred from the pressure changes. I have suggested some modified prose in a few places in the accompanying annotated manuscript, which are intended to make this distinction clearer,,,,, and hope that the authors would make a final attempt to make it clearer elsewhere as well.

Responses: We believe that the pore-pressure changes are the observations of SSE beneath the accretionary prism. But there are still some ambiguities in the source models of SSE due to a small number of pore-pressure observatories. Therefore, we would accept the reviewer's advice and modified the text accordingly.

The one concern raised by both myself and another reviewer, was that the authors do not present a serious consideration of other interpretations. They have added text in lines 54-56 and 60-64, but these new lines basically just note that different combinations of types of slow earthquakes occur in different places and times, and thus that "slow earthquakes have complicated source processes". I found Figure 2 striking evidence that at least would be consistent with the VLFs being consistent with the model presented in Gomberg et al. (2016), as it is definitely clear from Figure 2 that the VLFs could represent the envelope of clustered LFs. The diversity of focal mechanisms (see below) further supports this alternative interpretation. Acknowledging alternative models does not negate the one the authors clearly prefer, but a more objective presentation would demonstrate that other models have been considered and perhaps would even say why the one selected was preferred.

Responses: As recognized by the reviewer, we have added references to papers providing alternative interpretation. However, the reviewer seems to request more discussion on the model of Gomberg et al. (2016), hereafter G16. The reviewer may think that Fig. 2

shows evidence for the idea presented in G16, but it is actually counter evidence. In their model (Fig. 4 of G16), they demonstrated that the broadband spectra of tremors and VLFE, which has a sharp decrease of amplitude between 0.1-1 Hz. Such decrease cannot be observed in our observations (Fig. 2), and amplitude of velocity spectrum is rather constant for wide frequency range, which is well explained by the model of Ide (2008; 2010). The disproof of G16 is not in the scope of the present manuscript, and the detailed discussion on the broadband behavior of slow earthquakes like Fig. 2 will be provided as separate papers. Nevertheless, we agree that Fig. 2 is interesting and worth a little more discussion. Therefore, instead of the suggestion by the reviewer, we added sentences and a reference to a presentation in the 2018 AGU Fall Meeting (L. 89).

A related disturbing section of the paper is the discussion of the focal mechanisms, and fact that the authors present rather weak excuses for simply ignoring the mechanisms that do not agree with their model! The significant diversity of focal mechanisms is attributed to error and heterogeneity, yet some of the mechanisms are extremely different and yet somewhat spatially consistent (e.g. group of strike-slip events in darker blues). This would imply that many of the focal mechanisms are completely wrong, as it is hard to imagine what "heterogeneity in the plate interface" can explain them! The authors simply "assume that the sVLFEs generally represent low-angle thrust movement on the plate interface", despite evidence to the contrary!

Responses: In the Bootstrap error analysis of the VLFE source location, the average source location errors were 5 km in the horizontal and 2 km in the vertical direction. Therefore, most events were well located. Exceptions were several events that were deeper or outside the observation network, and we attributed the strike-slip events deeper than 10 km to the source mislocation in the previous manuscript.

In the revised manuscript, we further investigated errors of the focal mechanism, and the average errors of the plunge and azimuth of the principal axes were 16° and 29°, respectively, including the deeper strike-slip events (L. 280-286). Accordingly, we should not attribute these events due to structural heterogeneity, but these events would be considered to emerge from different source locations or processes than the other shallow thrust-type VLFEs. Further detailed studies are necessary to identify their sources. Aside from these minor activities, sVLFEs represent low-angle thrust movement on the plate interface, of which source location and mechanism were well constrained. We have revised the manuscript to clarify this point (L. 102-103).